# Do Attention and Memory Tasks Require the Same Lighting? A Study in University Classrooms

Carmen Llinares [1], Nuria Castilla [2,*] and Juan Luis Higuera-Trujillo [1]

[1] Institute for Research and Innovation in Bioengineering (i3B), Universitat Politècnica de València, 46022 València, Spain; cllinare@omp.upv.es (C.L.); jlhiguera@i3b.upv.es (J.L.H.-T.)

[2] Centro de Investigación de Tecnología de la Edificación (CITE), Universitat Politècnica de València, 46022 València, Spain

* Correspondence: ncastilla@csa.upv.es

**Abstract:** Lighting plays a fundamental role in learning spaces as it influences students' performance. Nowadays, new technologies and new teaching methods in higher education mean that very different visual tasks are performed in the classroom, so further research is necessary to identify what lighting is best suited to these new tasks. The objective of the study is to analyse the impact that variations in levels of illuminance and Correlated Colour Temperature (CCT) of classrooms have on the cognitive functions (attention and memory) of university students. The cognitive performance of 90 participants was evaluated based on attention and memory tasks. The participants had to view nine virtual classroom configurations, with three different illuminance settings (100 lx, 300 lx and 500 lx) and three CCTs (3000 K, 4000 K and 6500 K). The results showed that attention and memory tasks require different level of illumination. While attention improves with higher light levels, memory improves with lower light levels. Higher CCTs generate better performance in both attention and memory tasks. These results highlight the link between lighting and students' cognitive responses. This study and its methodology can be useful for architects and researchers as they establish lighting design guidelines capable of improving students' cognitive processes.

**Keywords:** classroom design; lighting; memory; attention; psychological responses; neuroarchitecture; human centric lighting

## 1. Introduction

Light plays a fundamental role in learning environments [1,2] as it affects students' performances [3] and academic achievements [4,5]. Light conditions influence task visibility, visual performance, comfort and the visual impressions of spaces, people and objects [6]. Moreover, light may change the way we perceive a space, how much we like it and how we feel [7].

New information and communication technologies and new learning practices have changed higher education. Researchers have called for learning spaces that are interactive, collaborative and that allow appropriate physical movement and social engagement among teachers and students [8]. Indeed, new technologies and new teaching methods mean that very different visual tasks are now performed in classrooms, so further research is necessary to identify what is the most appropriate lighting for these new tasks [9].

An important consideration is whether modifications made to educational spaces to adapt to new tasks, and the consequent changes made to lighting, can improve the energy efficiency of teaching centres. Illuminance level is one of the parameters of lighting sources with the greatest influence on energy expenditure. Illuminance, together with Correlative Colour Temperature (CCT), is widely recognised as a key interior lighting variable [10].

Traditionally, lower levels of lighting have been associated with decreased cognitive and academic performance [11–13], but recent research has investigated the influence of illuminance on cognitive performance and obtained conflicting and inconclusive results.

Two of the most important cognitive tasks associated with student learning are working memory and attention. Nevertheless, the results of studies that have analysed the effects of light on these two parameters have reached contradictory results.

In the case of attention, some studies have shown positive effects of bright light [14]. Smolders et al. [15], in a study with two illuminance levels (1000 lx vs. 200 lx at eye level, 4000 K) showed effects of illuminance on subjective alertness and vitality, sustained attention in tasks, and heart rate and heart rate variability. In another study, Smolders and de Kort [16] investigate if the effects of a higher illuminance level benefits individuals who suffer from mental fatigue. Like the previous study, they found that illuminance has a positive influence on attention. However, other studies such as Huiberts et al. [17] exposed each participant to three different light levels (1700 lux, 600 lux and 165 lux at eye level) for approximately one hour at the same time of day and did not find significant differences of illuminance on sustained attention. Other studies also found no significant differences [18,19]. Moreover, some such as Leichtfried et al. [20] revealed that high-intensity illumination (5000 vs. 400 lx at eye level) even impacted negatively on sustained attention. In the same way, the results of Öner et al. [21] and Leccese et al. [22] revealed that lower illuminances improved cognitive performance on a test requiring sustained attention.

The same could be considered in the case of working memory. While some studies show positive effects [17,23]; others found no significant differences, like J. Y. Park et al. [24]. They analysed the effects of two illuminance levels (700 and 150 lx) during a working memory task and observed that high illuminance led to significantly lower frontal EEG theta activity than did low illuminance. These differences persisted despite the fact that they did not find significant differences in task performance between illumination conditions. Furthermore, other studies suggest that participants even performed worse when they were exposed to higher illuminance levels (1000 vs. 200 lux) [23].

Research into CCT has also returned no conclusive results. Some authors have observed that higher CTT values generate better cognitive processing and performance [25]. Mills et al. [26] conducted a study within a shift-working call centre. They investigated the effect on well-being, functioning and work performance of fluorescent light sources with a high CCT (17 000 K) compared to a control group (2900 K). The data showed improved concentration for a group of employees working in high CCT. Viola et al. [27] exposed workers on two office floors to a change in lighting (17,000 K vs. 4000 K) for eight weeks. They evaluated daily alertness, mood, sleep quality, performance, mental effort, headaches and eye strain. Results showed that high CCT light improved all the tested parameters compared to normal white light. Keis et al. [28] in their experiment changed to blue-enriched white lighting (1400 k vs. 4000 K) to stimulate the circadian system of students at two schools. Results showed the beneficial effects of blue-enriched white light on students' performance.

Despite these results, other studies have concluded that CCT has no influence on attention or working memory. Smolders and de Kort [29] investigated the effects of CTT (2700 K vs. 6000 K, 500 lx on the desk) on individuals' experiences, performance and physiology during one hour of exposure in the morning versus afternoon. Results showed that CCT and time of day had no significant main or interaction effects on sustained attention. In the same vein, Vandewalle et al. [30] exposed participants to blue (470 nm) or green (550 nm) monochromatic light. Even though their results implied that monochromatic blue light could affect cognitive function, they did not find any difference in the performance in an auditory working memory task.

In addition, other studies have found that a combination of illuminance and CCT produces varied results. Ru et al. [31] analysed the influence of these two parameters and observed significant differences in the performance of attention and memory tasks with modified lighting levels, with better results for 1000 lx versus 100 lx, but not when the CCT was modified (3000 K vs. 6500 K).

Thus, in most cases the results are inconclusive. Perhaps this is due to the complexity of the measurement processes used since studies have varied in dose (illuminance levels),

light wavelength, timing and duration of exposure and task and task difficulty. Similarly, it should be considered that spaces themselves are difficult to analyse due to the diversity of variables that come into play and the limitations of the measurement systems used.

In turn, an important limitation when conducting lighting studies in real spaces is the economic cost of creating different scenarios. Virtual reality (VR) solves these problems as it offers the possibility of creating multiple scenarios that give the user the feeling of 'being there' [32]. VR generates psychological and neurophysiological responses like those generated by the physical environments represented [33]. VR has been validated for the execution of tasks in the specific case of university classrooms [34]. It has been used to assess attention problems [35,36] and memory [37]. In addition, it has been shown that VR is a more cost effective and efficient tool than real physical environments in the quantification of cognitive processes [38].

Taking into account the above aspects, the objective of the present study is to analyse the impact that variations in levels of illuminance and CCT in classrooms have on some cognitive functions (attention and memory) of university students. VR was used to generate different scenarios in a controlled manner. Different levels of illuminance and CCT were simulated, in which respondents perform cognitive tasks which can evaluate levels of attention and memory. Through this experience, and indirectly, the following questions are answered: (A) What levels of illuminance and CCT achieve better results in attention and memory tests? (B) Is it possible to perform cognitive tasks in a simulated environment?

## 2. Materials and Methods

The experimental methodology was a laboratory study. Comparisons were made of the effects of different classroom lighting configurations (through variations of illuminance and CCT), shown through VR, on the cognitive responses of experimental participants (attention and memory). Figure 1 shows a general outline of the study.

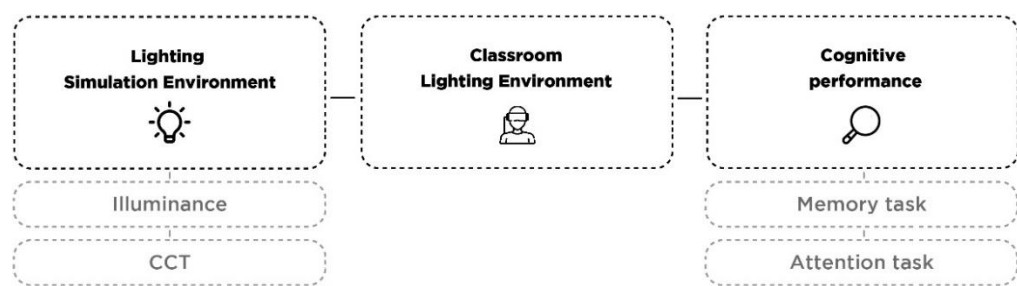

**Figure 1.** General outline of the study.

### 2.1. Stimuli

A representative classroom in the Higher Technical School of Building Engineering (ETSIE) at the Polytechnic University of Valencia, Spain, was virtualised. A classroom with no natural lighting was represented to avoid the interference of the variables natural and artificial lighting. A $3 \times 3$ matrix was configured, with 3 categories for the illuminance variable: 100 lx, 300 lx and 500 lx, and 3 categories for the CCT: 3000 K, 4000 K and 6500 K. Figure 2 shows the set of configurations. Each participant was exposed to all configurations, following a complete counterbalancing design.

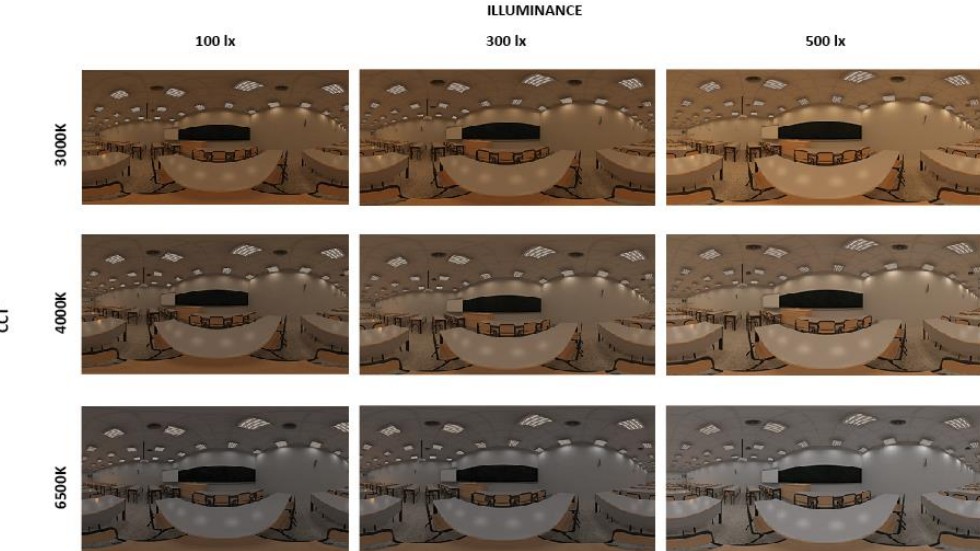

**Figure 2.** Lighting environmental simulations.

## 2.2. Environmental Simulation Set-Ups

The participants experienced each environment through VR simulations shown through a Head-Mounted Display (HMD). The simulations were developed through a process of modelling and rendering. The modelling was carried out using Rhinoceros [39]. The rendering process was carried out on the 3ds Max platform [40], using V-Ray [41]. It should be noted that irradiance map and light cache were used as the primary and secondary global illumination calculation engines, respectively; methods that allow the diffuse surface irradiance of objects in the scene to be calculated efficiently. VRayLightMeter, a tool for evaluating how a scene is lit like a real-life light meter used in photography, was used iteratively to adjust the scene until the working plane (located on the participant's virtual table) presented the three categories of lighting described in the previous paragraph. This allowed for photorealistic lighting, which controlled the two lighting parameters studied in this paper. The virtual implementation of the simulations was developed using Unity3D [42]. The HMD used was a HTC Vive device [43]. It has a resolution of $1080 \times 1200$ pixels per eye ($2160 \times 1200$ in total), with a field of view of $110°$ and a refresh rate of 90Hz. Figure 3 shows participants taking part in the experiment.

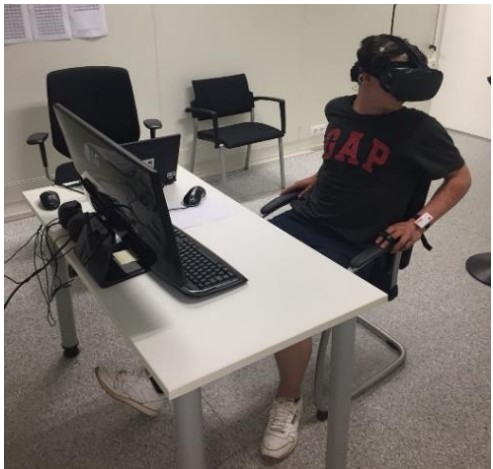 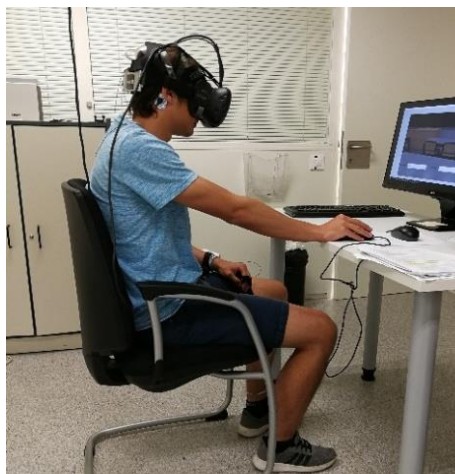

**Figure 3.** Participants during the classroom experiment.

### 2.3. Participants

A total of 90 participants took part in the study, 55% male and 45% female (average age = 22.75 years; σ = 2.348). Three inclusion criteria were established: (1) being a university student between 18 and 23 years (common ages among students studying for a university degree in Spain); (2) being Spanish (to avoid possible cultural effects); (3) having normal vision (to avoid problems with the use of VR displays) or corrected to normal through the use of contact lenses; and (4) not use substances (such as caffeine) or medication that could interfere with locomotion or performance.

### 2.4. Data Analysis

The sense of presence experienced by the participants in the simulated environments was assessed both by means of a questionnaire, and through their performances in attention and memory tasks. Table 1 displays the characteristics of the variables. In the design of the study, it was considered that task performance could depend on visual abilities and visual comfort. Therefore, auditory and non-visual tasks were chosen as the analysis focused on the effects of light on cognitive performance.

**Table 1.** Variables under study.

| **Presence** | | | |
|---|---|---|---|
| Objective | Measurement | Process | Metrics |
| Validate the degree to which a simulation can generate in the participant a response similar to that produced by the physical world | SUS questionnaire [30], which consists of six items evaluated on a Likert-type scale, from 1 to 7 | The participant assesses the sense of presence of each simulated environment analysed | • SUS-Total |
| **Attention task** | | | |
| Objective | Measurement | Process | Metrics |
| Quantify the degree of attention of the participants for each of the simulated environments | Attention task (similar to [32]), which consists of reacting as soon as possible to a specific auditory stimulus with a mouse click (objective) and avoiding doing so with other auditory stimuli (distractors) | The participant is subjected to a task with 8 objects and 32 distractors; the time between stimuli was 800 ms to 1600 ms. The participants had 750 ms to react to the stimuli, after which the episode was considered an error | • Attention-Time • Attention-Errors |
| **Memory task** | | | |
| Objective | Measurement | Process | Metrics |
| Quantify the memory level of the participants for each of the simulated environments | Memory task [31], which consists of remembering a set of related words | The participant listens to 3 audio clips of 15 words each and repeats them in a maximum time of 30 s | • Memory-Matches |

### 2.5. Procedure

Outliers were controlled by performing all experimental sessions following the same protocol. Figure 4 shows the different actions performed during the sessions. In addition, there was no daylight contribution during this experiment, and the background noise level, the ambient temperature, the arrangement of the furniture in the room and the devices used were kept constant for the whole sample.

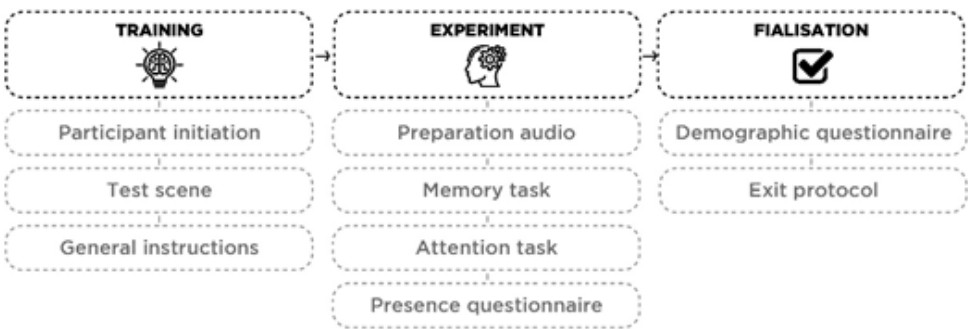

**Figure 4.** Experimental procedure.

The sessions were held during the university teaching timetable in Spain (8:00–21:00), so that the results would be representative of classroom activity. Given that the control of such a wide time slot would have required an excessively large sample, it was decided to carry out the sessions in a randomized way. This methodology is described by Kish [44] as a way of controlling an experiment by including variables randomly, on the basis that chance will generate equivalent distributions of the units in the variables under study. Thus, the remaining bias is smaller.

*2.6. Statistical Analysis*

The database was first anonymised, then the appropriate statistical analyses were carried out. Table 2 describes the analysis, statistical tests and expected results. IBM SPSS (v.17.0) was used. The analysis of level of sense of presence (Phase 1) was performed by summing the average level of six items that compose the SUS questionnaire. The analysis of the incidence of the different configurations of illuminance and CCT on memory (Phase 2) and attention (Phase 3) was carried out using statistical comparison techniques (parametric or non-parametric statistical tests depending on the distribution of the data of the variables). It was compared the response of the students to different levels of illuminance and CCT. This analysis made it possible to identify differences and, if they existed, which design configuration was associated with better or worse performance by the participant.

**Table 2.** Statistical tests.

| Phase | Analysis and Data Used | Statistical Treatment | Expected Result |
|---|---|---|---|
| Phase 1 Validation of the VR environment. | Analysis of level of sense of presence. SUS-Total | Descriptive analysis of means. | Sufficient level of sense of presence. |
| Phase 2 Memory analysis as a function of the classroom lighting environment. | Analysis of memory performance. Memory-Matches | ANOVA and Bonferroni's *post hoc* analysis (normally distributed data) for Memory-Successes, as a function of illuminance and CCT. | Significant differences in memory performance depending on classroom lighting (illuminance and CCT). Identification of the environmental simulations which gave the best and worst memory performances. |
| Phase 3 Attention analysis as a function of the classroom lighting environment. | Analysis of attention performance. Attention-Time Attention-Errors | ANOVA and Bonferroni's *post hoc* analysis (normally distributed data) for Attention-Time, as a function of illuminance and CCT. Kruskal-Wallis test and Mann Whitney's *post hoc* analysis (non-normally distributed data) for Attention-Errors, as a function of illuminance and CCT. | Significant differences in attention performance depending on classroom lighting (illuminance and CCT). Identification of the environmental simulations which gave the best and worst attention performances. |

## 3. Results

### 3.1. Validation of the VR Environment

The average levels of sense of presence experienced by each participant for each environmental simulation were obtained. The SUS questionnaire [45] was used for this purpose. It consists of six items, which are rated on a Likert scale from 1 (completely disagree) to 7 (completely agree). Taken together, the six items address three aspects of the sense of presence: being inside the simulation; regarding the simulation as real; and remembering the simulation as a place.

Table 3 shows the results provided by the SUS questionnaire for each item. The average of the set of items is 5.12 out of 7. The highest rated item (5.41) is "*I had a sense of "being there" in the classroom space*" and on the other hand the lowest rated item (4.81) is "*I think of the classroom space as a place similar to other places that I've been today*".

**Table 3.** Average level of sense of presence for each item of the SUS questionnaire. The results are presented using the mean and the standard deviation.

| Item | Illuminance | | | CCT | | |
|---|---|---|---|---|---|---|
| | 100 lx | 300 lx | 500 lx | 3000 K | 4000 K | 6500 K |
| 1. I had a sense of "being there" in the classroom space | 4.96 (1.380) | 5.62 (0.792) | 5.68 (1.129) | 5.25 (1.172) | 5.60 (1.103) | 5.36 (1.273) |
| 2. There were times during the experience when the classroom space was the reality for me | 4.68 (1.499) | 5.29 (1.084) | 5.20 (1.533) | 5.04 (1.467) | 5.12 (1.375) | 4.95 (1.440) |
| 3. The classroom space seems to me to be like somewhere that I visited before | 4.84 (1.677) | 6.00 (1.032) | 5.20 (1.778) | 5.13 (1.846) | 5.60 (1.366) | 5.18 (1.597) |
| 4. During the experience you felt you were in the classroom space | 4.60 (1.305) | 5.76 (0.928) | 4.96 (1.289) | 5.00 (1.364) | 5.32 (1.129) | 4.86 (1.335) |
| 5. I think of the classroom space as a place similar to other places that I've been today | 4.32 (1.839) | 5.29 (1.529) | 4.84 (1.816) | 5.00 (1.993) | 4.52 (1.663) | 4.86 (1.644) |
| 6. During the experience I often thought that I was really in the classroom space | 4.84 (2.047) | 4.62 (1.955) | 5.56 (1.368) | 4.92 (1.904) | 4.88 (1.874) | 5.32 (1.729) |

The sum of the six items of the SUS questionnaire provides the SUS-Total metric. Figure 5 shows the average value of this metric for the different simulations. They were considered to be sufficient, taking into account the results obtained by studies using similar technologies [46]. Thus, the simulations can be considered satisfactory.

### 3.2. Memory Analysis as a Function of the Classroom Lighting Environment

Memory was measured from the number of "successes" in the performance of the memory task. The higher the score, the higher the performance in the memory task. The statistical analysis applied depended on the normality of the data, which was examined using the Kolmogorov-Smirnov (KS) test. The illuminance and CCT results are described below.

#### 3.2.1. Illuminance

The average performances in the memory task for each illuminance level were obtained, and a subsequent search was made to identify any significant differences. Due to the normality of the data (KS, $p > 0.05$), an ANOVA was applied. Figure 6 shows a progressive reduction in memory test successes as illuminance increases. The ANOVA showed significant differences in memory test results as a function of illuminance ($p = 0.003$). The best performance in the memory task occurred in the case of the 100 lx illuminance.

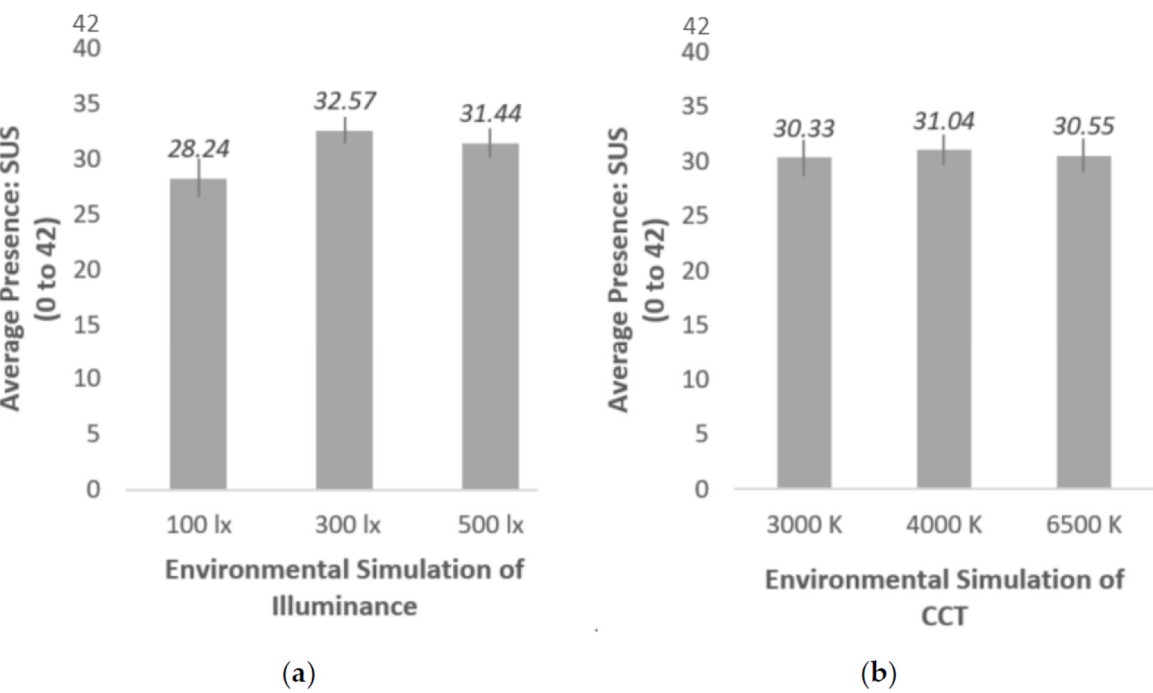

**Figure 5.** (**a**) Average level of sense of presence for environmental simulation of Illuminance (metric SUS-Total); (**b**) Average level of sense of presence for environmental simulation of CCT (metric SUS-Total).

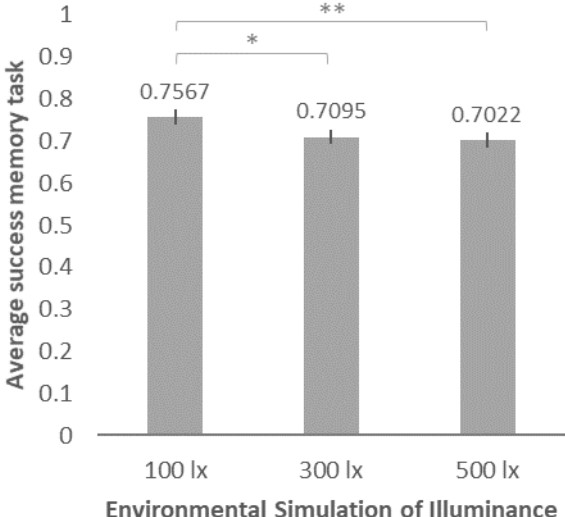

**Figure 6.** Average successes in memory task as a function of illuminance (metric Memory-Matches). The brackets indicate the comparisons and the asterisks the significance levels (* $p < 0.05$, ** $p < 0.01$).

The Bonferroni post hoc analysis showed that this difference occurred between the 100 lx environment and the other simulated environments, that is, 300 lx ($p = 0.037$) and 500 lx ($p = 0.003$).

### 3.2.2. CCT (Correlated Colour Temperature)

Figure 7 shows that memory test successes increase as CCT increases. The best scenario is 6500 K. The ANOVA test detected significant differences between the three scenarios ($p = 0.000$). The Bonferroni post hoc analysis showed that this difference occurred between the 6500 K environment and the other simulated environments, both at 4000 K ($p = 0.003$) and 3000 K ($p = 0.000$).

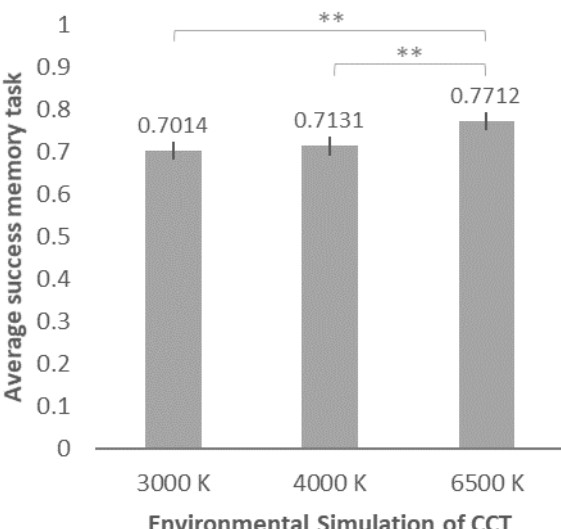

**Figure 7.** Average successes in memory tasks as a function of CCT (metric Memory-Matches). The brackets indicate the comparisons and the asterisks the significance levels (** $p < 0.01$).

### 3.3. Attention Analysis as a Function of the Classroom Lighting Environment

Attention was measured through two variables: (a) the reaction time in the attention task, that is, the longer the reaction time, the poorer the task performance; (b) errors made in the attention task. The illuminance and CCT results are described below.

#### 3.3.1. Illuminance

As Figure 8 shows, in both cases similar behaviour occurred. The illuminance that produced the worst performance in the attention task was 100 lx, with longer reaction times and more errors.

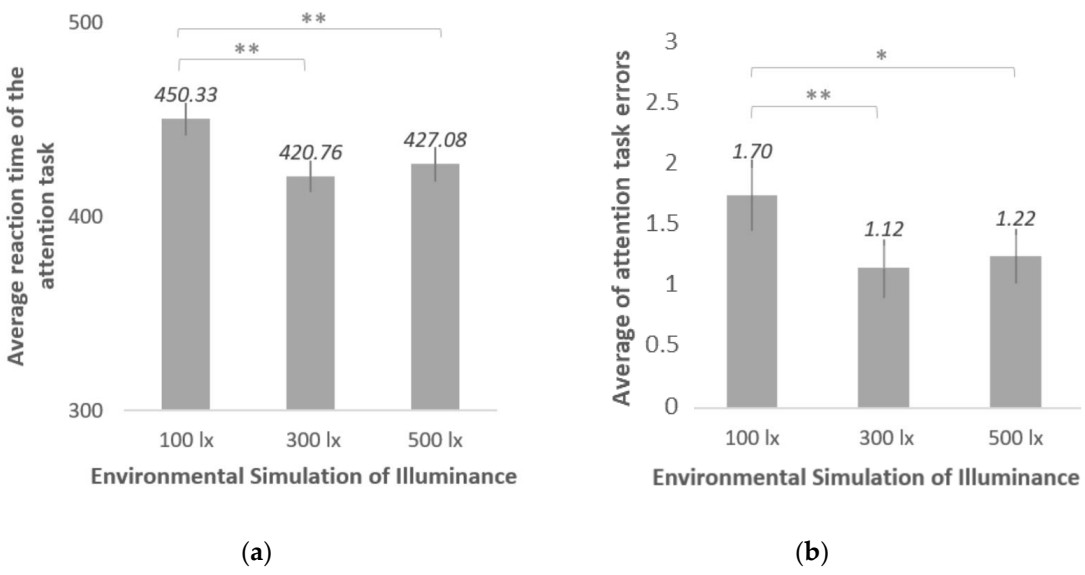

(**a**)                                                                                  (**b**)

**Figure 8.** (**a**) Average reaction time as a function of illuminance (metric Attention-Time); (**b**) average of errors made in the attention task as a function of illuminance (metric Attention-Errors). The brackets indicate the comparisons and the asterisks the significance levels (* $p < 0.05$, ** $p < 0.01$).

As the data follow a normal distribution, an ANOVA was applied to analyse the reaction time variable (KS, $p > 0.05$). This test showed significant differences in reaction time as a function of illuminance ($p = 0.000$). The Bonferroni post hoc test showed that this

difference occurred between the lowest illuminance, 100 lx, and the other two cases, 300 lx ($p = 0.000$) and 500 lx ($p = 0.000$).

A Kruskal-Wallis test was applied to the attention test errors as the data do not follow a normal distribution (KS, $p < 0.05$). This test showed significant differences in errors as a function of illuminance ($p = 0.009$). A post hoc analysis with Mann-Whitney tests showed that this difference occurred only between the cases of 100 lx and 300 lx ($p = 0.004$) and 100 lx and 500 lx ($p = 0.042$).

In both cases there were no significant differences between the 300 lx and 500 lx illuminances.

### 3.3.2. CCT (Correlated Colour Temperature)

Figure 9 shows the results for the CCT variable. For both variables the best result was obtained in the 6500 K case (shorter reaction times and fewer errors made in the attention task). The ANOVA (KS, $p > 0.05$) showed significant differences in reaction times in the attention task as a function of CCT ($p = 0.003$). The Bonferroni post hoc test identified significant differences when the CCT was reduced from 6500 K to 3000 K ($p = 0.006$) and to 4000 K ($p = 0.015$). The Kruskal-Wallis test (KS, $p < 0.05$) did not identify significant differences in the errors made in the attention task as a function of CCT.

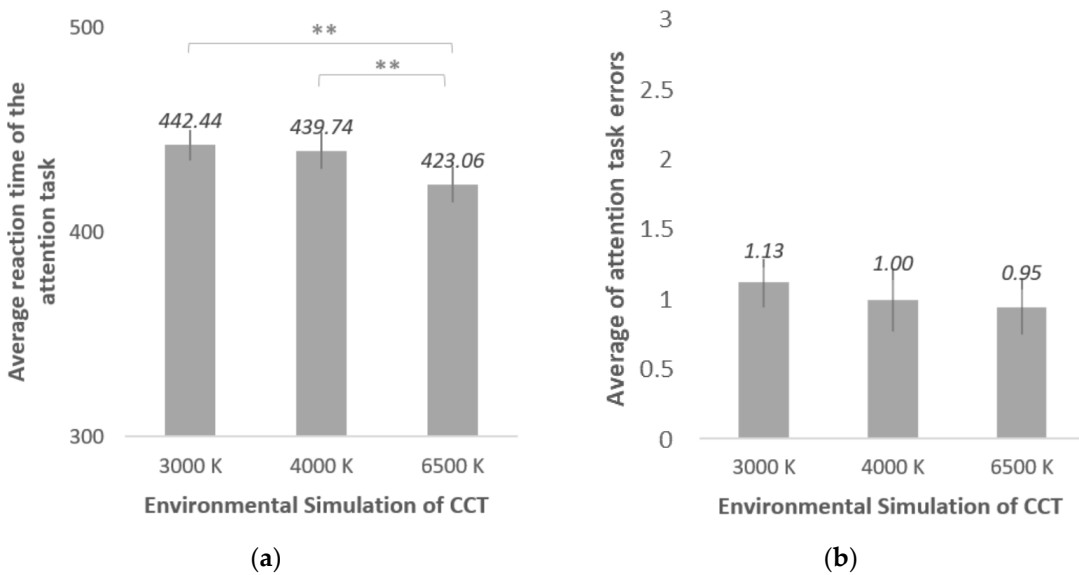

(a)                                (b)

**Figure 9.** (**a**) Average reaction time as a function of CCT (metric Attention-Time); (**b**) average of errors made in the attention task as a function of CCT (metric Attention-Errors). The brackets indicate the comparisons and the asterisks the significance levels (** $p < 0.01$).

### 4. Discussion and Conclusions

In the present study we analyse whether attention and memory tasks require similar lighting, in terms of illuminance and CCT. The performance of students in attention and memory tasks was analysed through VR while classroom lighting conditions were modified (illuminance and CCT) using an HMD VR tool.

The fundamental methodological contribution of this work lies in the use of VR for the simulation of different scenarios and the performance of tasks in these environments. In general, works undertaken on this subject have hitherto used physical spaces [23,28,31]. Through VR, researchers can control lighting characteristics without the interference of other environmental visual variables such as natural light or the observer's position with respect to the object being observed. It is important to note that, although virtual classrooms have been used, and validated, to measure levels of attention, memory and even student performance [47], very few studies have been undertaken to identify lighting design guidelines that optimise students' cognitive performance. This paper proposes that VR be

used to simulate lighting environments and be incorporated into the performance of tasks that can evaluate participants' cognitive performance (attention and memory).

The results of the present study allow interesting conclusions to be drawn about classrooms' lighting environments:

It was observed that lower levels of illumination (100 lx) generated different results in the performance of the attention and memory tests. On the one hand, the results of the attention test were poorer (longer reaction times and more errors) and, on the other hand, performance in the memory task improved. It was observed, therefore, that there was a significant change in the performance of both tasks when illuminance was reduced to 100 lx. These results are of great interest because they indicate that each cognitive function requires a different level of illuminance. In this sense, several authors have observed that a positive relationship exists between levels of attention and illuminance. For example, Smolders and Kort [16] observed improvements in attention levels and a reduction in mental fatigue when illuminance increased from 200 lx to 1000 lx. Other studies have found similar results, for example, when increasing from 5 lx to 1000 lx [48], and from 100 lx to 1000 lx [31]. However, other authors have observed that higher performance is not always associated with higher illuminance but depends on the complexity of the task [23]. These last authors observed that task complexity was a determining factor in the illuminance-performance relationship; they found that simple tasks were executed more successfully with less illuminance, while complex tasks were performed more successfully with increased lighting levels. This may explain the discrepancy found between the performance of the attention and memory task in this study. A result similar to one found in the present study was obtained by Smolders et al., that is, when illumination level was increased from 200 lx to 1000 lx, sustained attention levels improved, but performance in some memory tasks decreased, such that the best memory performance was produced with the lowest lighting level (200 lx).

These results demonstrate it is possible to adapt lighting levels to optimise performance in the different cognitive tasks that students undertake in classrooms. This, given the possibilities offered by dynamic lighting and lighting sources such as LEDs, can have significant repercussions for educational centres' energy expenditure, and help improve student performance. It would be interesting, therefore, to extend the present study by assessing performance in different tasks (related to attention, memory and motivation) to contrast the results. However, it is essential that, to make valid comparisons with the results obtained in other studies, that similar lighting levels, light wavelengths, timing and duration of exposure, task types and difficulty are analysed.

It was observed that higher CCTs generated significantly better results both in the attention and memory tasks. The best performances in both tasks being obtained in the 6500 K scenario, that is, when increasing from 3000 K and 4000 K to 6500 K. Increasing from 3000 K to 4000 K generated performance improvement in both tests, but not significantly. This result is consistent with most of the studies carried out, that is, they have found that higher levels of CCT are associated with improved concentration [26–28].

Finally, two limitations of the study should be highlighted. On the one hand, the time of day was not controlled for in this study. This would have required a very large sample of participants, given that teaching activities at Spanish universities cover a very wide time slot (8:00–21:00). In future studies, it would be interesting to study the different time intervals in more detail to verify that the presented results are consistent. On the other hand, the possibility of unexplored synergistic effects between lighting variables and classroom design should be discussed. In the present study, to avoid this effect, all variables were kept constant, apart from illuminance and CCT. It would be interesting to see if the results of the study are maintained when other lighting design variables were modified, such as uniformity, and to analyse the joint effect of the combination of these variables.

**Author Contributions:** Conceptualization, C.L., N.C., and J.L.H.-T.; Data curation, C.L., and N.C.; Formal analysis, C.L.; Funding acquisition, C.L.; Investigation, C.L., N.C., and J.L.H.-T.; Methodology, C.L., N.C., and J.L.H.-T.; Project administration, C.L.; Resources, C.L.; Software, J.L.H.-T.; Supervision, C.L.; Validation, C.L.; Visualization, J.L.H.-T.; Writing—original draft, C.L., and N.C.; Writing—review & editing, C.L., N.C., and J.L.H.-T. All authors have read and agreed to the published version of the manuscript.

**Funding:** This work was supported by the Ministerio de Economía, Industria y Competitividad of Spain (Project BIA2017-86157-R; and PRE2018-084051).

**Institutional Review Board Statement:** The study was conducted according to the guidelines of the Declaration of Helsinki and approved by the Institutional Review Board Review Board of the Universitat Politècnica de València (P1_25_07_18; 25 July 2018).

**Informed Consent Statement:** Informed consent was obtained from all subjects involved in the study.

**Conflicts of Interest:** The authors declare that they have no known competing financial interests or personal relationships that could have appeared to influence the work reported in this paper.

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
