# Peer review of "Do Attention and Memory Tasks Require the Same Lighting? A Study in University Classrooms"

_sustainability, doi:10.3390/su13158374_

Round 1

Reviewer 1 Report

The topic dealt with in the manuscript is currently much debated in the scientific community and of undoubted interest. The study is approached with innovative tools and presents interesting ideas, therefore it deserves to be published in Sustainability.
However, some minor changes are required before full acceptance.
The main aspect that needs to be changed is the validation of the virtual environment.
It is essential to know if the virtual environment and the real one are perceived in the same way by users, otherwise any observation obtained using the virtual environment would be meaningless. For the aforementioned reason, section 3.1 cannot be treated as briefly as the authors did. All information used for validation must be indicated and the validation itself must be supported by solid considerations.
Another aspect to change is the introduction. Authors should not limit themselves to listing references to support some generic comments, but they should be a little more specific, mentioning the results obtained by other research groups and the characteristics that make their study and their results different and original compared to the previous one.
Also some further inherent studies should be mentioned (for example DOI:10.1177/0143624419894441, DOI:10.1177/1420326X19864414, DOI:10.1016/j.jenvp.2014.03.005).

Author Response

We are grateful to the reviewers for their valuable comments. We appreciate their time and effort in improving the manuscript. The manuscript has been updated using those comments and also, we have added new paragraphs to enhance the document.

Please, consult the attached file.

Reviewer 2 Report

The article has some merit but some more points need to be analyzed.

The factor of time of day should have been discussed. Do the results stand the same on whether the experiment is made in the morning, afternoon or evening? Does the current state of the circadian rhythm play a role? Should morning classed and evening classes be lit in the same way?

More technical details should be given on how Illuminance and CCT simulated correspond to actual values on the display. Has a color calibration check been made on the VR display? Is user validation enough to accept similarity with the physical world? Does the display perform equally well in all CCT's simulated?

Sample text if left in the beginning of section 4.

Author Response

(The authors gave the same response as above.)

Reviewer 3 Report

This work study the impact that variations in levels of illuminance and correlated color temperature (CCT) of classrooms have on the cognitive functions (attention and memory) of university students. The cognitive performance of 90 participants was evaluated based on attention and memory tasks. The subjects had to view 9 virtual classroom configurations, with 3 different illuminance settings. These results highlight the link between lighting and students' cognitive responses.

The topic of lighting associated to attention and memory is meaningful and relevant for an international audience. The manuscript presents a significant scientific contribution however, in general some clarifications and improvements must be made:

Capitalize the initial letter in words related to acronyms for example page 1 line 16: correlated color temperature (CCT).

Reference the web pages with numbers in the text and include the link in References

Follow instructions for authors (references in text related to in square brackets.

In 2. Material and Methods: explain data analysis

In 2.4 Describe the measurement/measure and its procedure. Does the time or hours of the day was it done influence in the results? Describes the variables that can produce errors.

In table 1 differentiate the first line of text from the rest (objective, measure.) by means of a line and/or or bold.

In figure 4 on the Y axis indicate the data of the maximum number (42).

Author Response

(The authors gave the same response as above.)

Round 2

Reviewer 2 Report

Authors addresses issues.